# Online Error-Related Potential Classification via Predictive Coding and Reservoir Computing

**Mac O'Brien**
Department of Computer Science and Technology
Tsinghua University
amw25@mails.tsinghua.edu.cn

**Worawate Ausawalaithong**
Department of Computer Science and Technology
Tsinghua University
ma-xy25@mails.tsinghua.edu.cn

## Abstract

In this study, we introduce a method for online time series classification using predictive coding and reservoir computing. We apply our method to single trial ErrP classification using the ErrP-HRI dataset, which exhibits significant intra-subject variability. Our approach replaces the standard regularised linear discriminant analysis with a predictive coding reservoir, enabling rapid test-time adaptation with limited data. We hope to demonstrate the superiority of predictive-coding and reservoir computing for data-scarce and highly variable time series classification as compared to traditional methods.

## 1 Introduction

Closed-loop human–robot interaction (HRI) using error-related potentials (ErrPs) imposes strict requirements on a decoder: it must classify noisy, single-trial EEG within hundreds of milliseconds, adapt on-line to shifting signal distributions, and work reliably with only a handful of labelled examples per user. A model that can rapidly adjust to a new user or to a different robot stimulus—without full re-calibration—would substantially improve the practicality of neuro-based robot validation. However, the established baseline for single-trial ErrP classification, regularised linear discriminant analysis (rLDA) [1], although fast and data-efficient, is highly sensitive to exactly the distribution shifts that arise when moving from cursor to robot feedback; accuracy drops by more than 20 percentage points [2]. Deep-learning models such as EEGNet [3] can learn from limited data but typically require batch training and lack mechanisms for rapid on-line adaptation. This motivates a time-series classifier that is simultaneously data-efficient, robust to non-stationarities, and able to handle the low signal-to-noise ratio characteristic of real-world EEG.

### 1.1 Gaps Identified

Standard methods for ErrP classification require bespoke subject- and condition-specific models and perform poorly on cross-condition evaluation. This suggests that a more flexible approach is required for real world applications, which cannot be expected to adhere to strictly controlled laboratory conditions. While compact deep learning models such as EEGNet can operate with limited data, they typically require batch training and lack mechanisms for rapid online adaptation to non-stationarities during deployment. Existing decoders, once trained, remain fixed and cannot adjust to the distribution shifts inherent in EEG recordings without full re-calibration.

## 1.2 Novelty / Contribution

We are the first to apply a predictive coding-driven reservoir computing approach to ErrP classification. Our key contributions are: (1) a biologically plausible predictive coding reservoir architecture that explicitly models temporal dynamics and uncertainty for noisy EEG time series; (2) an online adaptation mechanism using FORCE learning that updates readout weights with single-trial latency, enabling rapid test-time calibration without full retraining; and (3) a systematic evaluation on the ErrP-HRI dataset.

## 2 Background

### 2.1 Error-Related Potentials (ErrPs)

Error-related potentials (ErrPs) are observable changes in the electrical activity of the brain that follow observed deviations from expected behavior. They are used in brain-computer-interfaces (BCI) as a way to automatically detect and correct erroneous behavior of the BCI. Beyond traditional BCI control, ErrPs have emerged as a powerful implicit feedback channel in human-robot interaction (HRI). When integrated into interactive robotic systems, ErrP-based signals enable continuous, trial-and-error adaptation through reinforcement learning, allowing robots to align their policies with human expectations and preferences [2].

### 2.2 Reservoir Computing

Reservoir computing is a specific scenario of recurrent neural network that consists of three layers: input, reservoir, and output. The input layer maps input to a high-dimensional reservoir state. The reservoir layer combines input and previous reservoir state to create a new reservoir state. This layer is often a pool of recurrently connected neurons, hence the name reservoir. The output layer maps reservoir state to output. This layer is often a simple linear regression. In reservoir computing, only the output is trained while the input and reservoir layers are kept fixed. As such, it requires significantly less computation, less data, and less training time than traditional gradient descent based neural networks. Reservoir computing is a unified framework of two methods that were introduced around the same time: echo state network (ESN) [4], which utilizes artificial neurons, and liquid state machine (LSM) [5], which utilizes biologically-inspired spiking neurons.

### 2.3 Predictive Coding in Reservoir Computing

Drawing inspiration from the manner in which the brain is thought to process sensory information, Katori proposed a neural network model based on predictive coding and reservoir computing (PCRC) [6]. In this framework, the reservoir generates predictions of future inputs, and the resulting prediction errors drive both inference and learning. The connection between the reservoir and the prediction layer is trained using the first-order reduced and controlled error (FORCE) algorithm [7]. By combining the high-dimensional temporal representation of reservoir computing with the error-minimizing dynamics of predictive coding, PCRC offers a computationally lightweight architecture capable of rapid adaptation.

## 3 Related Works / Existing Methods

### 3.1 Regularised Linear Discriminant Analysis

The current baseline for single-trial ErrP classification is rLDA [1]. Ehrlich and Cheng [2] demonstrated that rLDA achieves $90.6 \pm 3.9\%$ accuracy on cursor feedback but drops to $69.0 \pm 7.9\%$ on robot feedback, with cross-condition transfer degrading by over 20 percentage points. This fragility stems from rLDA's assumption of stable feature distributions, which is violated when stimulus type changes.

## 3.2 Deep Learning Approaches

EEGNet [3] is a compact convolutional network designed specifically for limited-data EEG classification. It has been validated on ERN (a component of ErrP) and achieves competitive performance across multiple BCI paradigms [3]. While EEGNet can learn from limited data, it typically requires batch training and does not support the rapid online adaptation needed for closed-loop HRI scenarios. Deep Convolutional Neural Network is another popular choice for ErrP classification [8, 9, 10, 11], however, there remains a significant lack of literature addressing their performance in practical, online scenarios [12]. Recent LSTM-based online ErrP decoders have achieved ∼73% accuracy with minimal training resources [13], but still require offline training on balanced datasets and lack test-time adaptation.

## 3.3 Reservoir Computing for EEG

Reservoir computing has been successfully applied to EEG-based classification tasks due to its low training cost and natural temporal processing. ESNs have been used for mental task classification in BCI, achieving two-task accuracies up to 95% with portable EEG systems [14]. ESNs have also been applied to epileptic seizure detection, where reservoir computing with Bayesian relevance regression achieved state-of-the-art performance with detection delays under one second on rat intracranial EEG [15]. However, these applications do not address the specific challenges of cross-condition transfer and online adaptation that characterize ErrP decoding in HRI.

# 4 Challenges

The ErrP-HRI dataset presents specific challenges: (1) The signal-to-noise ratio is extremely low in general, and especially pronounced in the robot condition compared to the cursor condition; (2) There is significant distribution shift between conditions; (3) The number of trials per subject is limited, preventing the use of large deep learning models.

# 5 Objectives

Our primary objective is to evaluate the efficacy of a reservoir-based predictive coding model for online ErrP classification on the ErrP-HRI dataset. We aim to match or exceed the within-session performance of the original rLDA baseline while significantly improving cross-condition robustness.

Secondarily, we will conduct ablation studies to isolate the contributions of the predictive coding layer and the online adaptation mechanism. We will evaluate computational efficiency to confirm that the model meets real-time constraints for closed-loop HRI.

# 6 Dataset

We will be using the ErrP-HRI dataset, which contains EEG signals for detecting error-related potential (ErrPs) in human subjects observing robots (or cursors) [2]. When the human observer perceives an incorrect action performed by the robot, they exhibit a characteristic waveform measurable in the fronto-central ad fronto-parietal sites. Eleven healthy participants (6 male, 5 female; age $29.4 \pm 7.4$ years) participated in two variations of an instruction and response task: they would give instructions for one of three possible directions via key press, which either a robot's head or a cursor would move towards (considered correct) or away from (incorrect). Each session consisted of 10 blocks of 50 trials, yielding 500 trials per participant per condition, of which approximately 65% were non-error trials and 35% were machine-error trials (error probability alternated between $p_{err} = 20\%$ and $p_{err} = 50\%$ across blocks). On average, each subject contributed approximately 325 non-error and 159 machine-error epochs per condition.

The data was recorded using a 32-channel active EEG system at 1000 Hz; downsampled to 256 Hz.

We chose the dataset primarily for two reasons: First, there is very high variability due to the non-stationary nature of the EEG data, even within a single-subject which motivates an approach that is robust to such a shift in distribution. Second, its cross-condition design (cursor vs robot feedback)

introduces another layer of distribution shift, making this an ideal benchmark for adaptation and generalization.

# 7 Proposed Methodology

## 7.1 Overall Pipeline

The proposed pipeline consists of four stages: (1) band-pass filtering, resampling, and epoching of raw EEG; (2) mapping of temporal features into a high-dimensional reservoir state space; (3) predictive coding-driven readout that generates class predictions and computes prediction errors; and (4) online weight adaptation using FORCE learning to update the readout layer without retraining the reservoir. This architecture enables the model to leverage fixed, randomly initialized reservoir dynamics while retaining flexibility through a rapidly adaptable readout.

## 7.2 Preprocessing

Following the baseline methodology drawn from S. K. Ehrlich and G. Cheng, the EEG data will be band-pass filtered (1–20 Hz) to remove high-frequency noise and DC offsets [2]. We then resample the data to 64 Hz, which is sufficient for ErrP analysis according to Nyquist rule since we are mainly interested in low frequencies component. This not only makes the processing faster, but also reduces overfitting and complexities from the curse of dimensionality [12]. We extract epochs from $-500$ ms to $+1500$ ms relative to feedback onset and apply baseline correction using the $-200$ ms to $0$ ms window. Consistent with the baseline, we use 27 EEG channels and discard EOG and peripheral channels.

## 7.3 Echo State Network

Our method utilizes a variation of reservoir computer called Echo state network (ESN), which is based on recurrent neural network and operates on discrete time. The reservoir state update in an ESN, which follows the standard ESN update rule, given in Appendix A.

## 7.4 Predictive Coding

The readout layer operates as a predictive coding network. It generates a prediction $\hat{y}[t]$ from the current reservoir state $x[t]$ via a linear mapping with weights $\theta$. The prediction error $e[t] = y[t] - \hat{y}[t]$ is computed with respect to the supervised target. This error drives both inference and learning: the readout weights are updated to minimize the squared prediction error. By framing classification as a prediction-error minimization problem, the model naturally handles noisy observations and can incorporate prior expectations about class probabilities.

## 7.5 Online Adaptation

We implement online adaptation using the FORCE (First-Order Reduced and Controlled Error) algorithm [7], which is well-suited for reservoir computing with output feedback. FORCE uses recursive least-squares to update output weights incrementally; the exact update equations are given in Appendix B. This approach requires only milliseconds of computation per trial, satisfies real-time constraints for closed-loop HRI, and allows the decoder to adapt to gradual non-stationarities or sudden condition shifts without re-calibration from scratch.

# 8 Progress So Far

We have implemented the core predictive coding reservoir model described in Section 7 and validated it on two forecasting benchmarks. The pipeline shows promise in time series forecasting, but is quite sensitive to hyper-parameter tuning and seems to smooth over fine-grained details. In Table 1, we provide our planned schedule to complete the project. For all phases, both group members will collaborate equally.

| Week | Dates | Phase | Tasks |
|---|---|---|---|
| 1 | May 4 - May 10 | Research and Dataset Preparation | Finalize preprocessing pipeline and implement baselines |
| 2 | May 11 - May 17 | Pipeline and Parameter Tuning | Implement our methods |
| 3 | May 18 - May 24 | Ablation and Experimentation | Run all experiments + ablation tests |
| 4 | May 25 - May 31 | Analysis | Collect all results and analyze them |
| 5 | June 1 - June 10 | Report and Presentation | Finalize written report and project presentation |

Table 1: Project timeline and task allocation.

## A  ESN Update Rule

The ESN update is governed by the following equation:

$$x[t+1] = (1-\alpha)x[t] + \alpha f(W_{in}u[t] + Wx[t] + b), \tag{1}$$

where $u[t]$ is the input signal, $x[t]$ is the reservoir state, $\alpha$ is the leaking rate, $f$ is the $\tanh$ non-linearity, and $b$ is the bias. The spectral radius of $W$ is scaled below unity to ensure the echo state property, yielding a stable response and fading memory of past inputs.

## B  FORCE Recursive Least-Squares Updates

The FORCE algorithm updates the output weight vector $W^{out}$ and the inverse correlation matrix $P$ at each time step as follows:

$$P(t+1) = P(t) - \frac{P(t)\,x(t)\,x(t)^\top P(t)}{1 + x(t)^\top P(t)\,x(t)}, \tag{2}$$

$$W^{out}(t+1) = W^{out}(t) - \frac{e(t)\,x(t)^\top P(t)}{1 + x(t)^\top P(t)\,x(t)}, \tag{3}$$

where $x(t)$ is the reservoir state vector, $e(t) = y(t) - \hat{y}(t)$ is the prediction error, and $P(0) = I/\alpha$ (with $\alpha$ a small positive constant). These equations implement an online recursive least-squares solver that minimizes the cumulative squared prediction error while being computationally efficient.

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
