# OpenReview forum: "Online Error‑Related Potential Classification via Predictive Coding and Reservoir Computing"
_tsinghua.edu.cn/THU/2026/Spring/ANM — THU 2026 Spring ANM Submission_

### Official Review · Reviewer_APTx · 2026-05-13

**Rating:** 8
**Confidence:** 3

**Summary:**

Current methods for ErrP classification such as rLDA are fast but highly sensitive to distribution shifts, dropping accuracy by over 20% when moving between conditions, while deep learning models like EEGNet lack mechanisms for rapid online adaptation. To address these limitations, this project proposes a framework combining predictive coding and reservoir computing (PCRC) to classify error-related potentials (ErrPs) from EEG signals in real time.

The pipeline starts by preprocessing raw EEG signals through band-pass filtering, resampling to 64Hz, and extracting one-second epochs around feedback events. The cleaned signal is then mapped into a high-dimensional reservoir state space using an Echo State Network (ESN), which captures temporal dynamics without requiring full retraining. A predictive coding readout layer then generates class predictions by minimizing prediction errors, and online adaptation is achieved through the FORCE algorithm, which updates readout weights incrementally after each trial without retraining the entire model. The model is evaluated on the ErrP-HRI dataset across both cursor and robot feedback conditions to test cross-condition generalization.

**Strengths:**

- Online adaptation: FORCE learning updates the model after every single trial in milliseconds, allowing real-time adjustment to distribution shifts without full retraining
- Data efficiency: Reservoir computing requires significantly less data and training time than deep learning models, making it well suited for limited EEG data scenarios
- Biologically plausible architecture: The predictive coding framework mirrors how the brain processes sensory information, providing a theoretically grounded approach to noisy EEG classification

**Weaknesses:**

- Hyperparameter sensitivity: The model is highly sensitive to hyperparameter tuning, as acknowledged by the authors themselves, which raises concerns about reproducibility and stability across different subjects
- Unclear adaptation speed: The authors claim single-trial latency adaptation through FORCE learning, but never report how many trials are actually needed before the model stabilizes to a reliable accuracy.
- Unaddressed class imbalance: The dataset contains roughly 65% non-error and 35% error trials, yet the paper does not discuss how this imbalance affects the predictive coding readout or whether any correction strategy was applied. This raises concerns about whether the reported accuracy is genuinely balanced or simply biased toward the majority class.

**Questions:**

- How many trials does FORCE actually need before it adapts reliably?
- How does the model handle the class imbalance?

---

### Official Review · Reviewer_ScwF · 2026-05-14

**Rating:** 7
**Confidence:** 3

**Summary:**

The authors propose an online time series classification method for single-trial Error-Related Potential (ErrP) detection in human-robot interaction. The method combines predictive coding with reservoir computing, using an Echo State Network to map EEG signals into a high-dimensional temporal state space and FORCE learning to update readout weights online. The proposal targets the ErrP-HRI dataset, where the main challenge is distribution shift between cursor and robot feedback conditions. The goal is to match or improve rLDA baseline performance while improving cross-condition robustness and online adaptation.

**Strengths:**

- Strong Motivation: The proposal clearly explains why ErrP classification is difficult: EEG signals are noisy, single-trial decoding is low-SNR, and the distribution shifts between cursor and robot feedback. The motivation for online adaptation is convincing.
- Good Connection to Time Series Modeling: Reservoir computing is a natural fit for temporal EEG data because it can model dynamics without expensive full-network training.
- Clear Novelty Claim: The authors clearly state that applying predictive-coding reservoir computing to ErrP classification is their main contribution. The combination of predictive coding, reservoir states, and FORCE learning is interesting and biologically motivated.
- Useful Mathematical Detail: The proposal includes the ESN update rule and FORCE recursive least-squares equations, which makes the core model more concrete than many high-level proposals.
- Relevant Dataset Choice: The ErrP-HRI dataset is well matched to the stated problem because it contains both cursor and robot feedback conditions, creating a natural cross-condition generalization benchmark.

**Weaknesses:**

- Online Adaptation Assumption Is Unclear: FORCE learning updates the readout using prediction error, but in a real online ErrP decoder it is unclear where the correct label comes from at test time. If the model updates using test labels, the evaluation may become unrealistic or overly optimistic. The proposal needs to clearly define whether labels are available immediately, delayed, inferred from robot correctness, or not available at all.
- Evaluation Protocol Lacks Detail: The proposal says it will evaluate cross-condition robustness, but the exact train/test setup is not fully specified. For example, it is unclear whether the main experiment is train on cursor and test on robot, train on robot and test on cursor, subject-specific adaptation, cross-subject generalization, or some combination.
- Missing Visualization: A system diagram showing preprocessing, reservoir state update, predictive coding readout, FORCE update, and final classification would help clarify the method.

**Questions:**

During online adaptation, what labels or feedback signals are available to compute the FORCE update?

---

### Official Review · Reviewer_9teJ · 2026-05-14

**Rating:** 7
**Confidence:** 3

**Summary:**

The proposal uses an Echo State Network with a predictive coding readout, updated online via FORCE learning, to classify single-trial ErrPs and adapt rapidly to distribution shifts.

**Strengths:**

•	Novel combination of reservoir computing and predictive coding for EEG decoding.
•	Directly addresses non-stationarity and cross-condition shifts, both being relevant time-series challenges.
•	Real-time compatible through computationally light FORCE updates.

**Weaknesses:**

•	Unclear how per-time-step predictions are aggregated into a trial-level classification decision, leaving the overall classifier undefined.
•	Predictive coding reduces to linear readout with error minimization, there is no hierarchical prediction or generative model, so novelty over a standard ESN with online RLS is questionable.
•	Online adaptation requires labelled target-condition trials but the number, timing, and protocol are unspecified

**Questions:**

1.	Which online-adaptive classifiers will be included as baselines?
2.	How will hyperparameters be selected given small per-subject sample sizes and cross-condition transfers and how do you plan to address the mentioned issue of the pipeline being quite sensitive to hyper-parameter tuning and seeming to smooth over fine-grained details?

---

### Official Review · Reviewer_VdiF · 2026-05-15

**Rating:** 7
**Confidence:** 3

**Summary:**

The paper proposes a novel approach to online ErrP classification by combining predictive coding and reservoir computing, specifically using an echo state network with a FORCE-trained readout layer. The goal is to address two well-known limitations of the current rLDA baseline: poor cross-condition transfer (cursor vs. robot feedback) and inability to adapt online to distribution shifts. The proposal is evaluated against the ErrP-HRI dataset, chosen for its high intra-subject variability and cross-condition design.

**Strengths:**

The problem framing is sharp. The authors identify a concrete and well-documented failure mode of rLDA, the 20+ point accuracy drop under condition shift and build their entire proposal around fixing it.
The choice of FORCE learning for online adaptation is well-motivated. The argument that recursive least-squares updates can operate at single-trial latency without retraining the reservoir is technically sound and directly relevant to the closed-loop HRI constraint.
The dataset choice is appropriate and well-justified. The ErrP-HRI dataset's cross-condition design is genuinely a good stress test for the proposed adaptation mechanism.

**Weaknesses:**

The novelty claim needs tightening. "First to apply PCRC to ErrP classification" is technically accurate but narrow, the architecture is directly from Katori (2018) and ESNs have already been applied to EEG. The authors should be more upfront that the contribution is primarily an application and adaptation of existing components to a specific underexplored problem, which is still a valid contribution.
The proposed ablation studies are mentioned but not well-defined. Which exact components will be ablated, under what conditions, and what metric will be used to isolate the predictive coding contribution from the reservoir dynamics alone? A cleaner ablation plan would strengthen confidence that the proposal is evaluable.
The class imbalance (~65/35) is never mentioned despite being a known challenge in ErrP classification.

**Questions:**

What is the exact experimental protocol for measuring cross-condition transfer?

How do you plan to handle hyperparameter selection without leaking information from the test condition? Given the small per-subject trial count, this is non-trivial.

Have you considered a pure ESN baseline (without predictive coding) as an intermediate ablation point between rLDA and the full PCRC model?

---

### Official Review · Reviewer_vUw1 · 2026-05-15

**Rating:** 7
**Confidence:** 4

**Summary:**

This proposal targets online error-related potential classification using a predictive coding reservoir architecture. The authors combine an echo state network with a predictive coding readout and FORCE learning to replace standard rLDA baselines, focusing on the ErrP-HRI dataset where cross-condition shifts cause existing methods to degrade by over 20 percentage points. A four-stage pipeline is outlined with preliminary forecasting results.

**Strengths:**

The authors have identified a clear and practical gap in brain-computer interface research. They note that rLDA accuracy drops sharply from 90.6% to 69.0% when moving from cursor to robot conditions, and they correctly argue that existing deep learning approaches fail here due to batch training requirements and lack of online adaptation. The methodological combination of predictive coding, reservoir computing, and FORCE learning is well suited to this setting. Predictive coding offers a principled way to model temporal dynamics and uncertainty in noisy EEG, while reservoir computing avoids expensive backpropagation through time. FORCE learning is a sensible choice for single-trial updates in data-scarce online settings. The ErrP-HRI dataset is an excellent choice with public baselines and realistic constraints, and the project scope feels appropriately bounded for a course timeline.

**Weaknesses:**

The current validation is limited to generic forecasting benchmarks rather than actual EEG data, which is problematic because ErrP classification involves non-stationary, extremely low-SNR signals that forecasting tasks do not replicate. The authors should prioritize even a single-subject pilot to surface real failure modes early. The predictive coding readout also lacks mathematical clarity; the proposal does not explain how computing hierarchical prediction errors and minimizing free energy translates concretely into the readout layer of an echo state network, making it difficult to assess whether this is genuinely novel or a standard ESN with relabeled terminology. The downsampling from 1000 Hz to 64 Hz risks blurring early transients like the ERN/Ne peak, and the authors should justify this or compare against 128 Hz. The 65/35 class imbalance is mentioned but not addressed in training or evaluation metrics, and with only 11 participants, the statistical testing strategy should be specified. Finally, the online adaptation protocol remains unclear: it is not stated whether FORCE updates happen continuously, during brief calibration windows, or as few-shot initialization, and no safeguards against catastrophic forgetting are described.

**Questions:**

What is the exact mathematical distinction between your predictive coding readout and a standard FORCE-trained linear readout?

When can you run a pilot on real ErrP data, and which hyperparameters prove most sensitive?

How will you handle the 65/35 class imbalance during FORCE learning?

Which 5 of the 32 EEG channels are excluded, and why?

Is FORCE adaptation continuous, intermittent, or few-shot initialization?

---

### Official Review · Reviewer_czJV · 2026-05-16

**Rating:** 9
**Confidence:** 3

**Summary:**

The authors propose a novel framework for online Error-Related Potential (ErrP) classification using a combination of Predictive Coding (PC) and Reservoir Computing (RC). The primary innovation is the replacement of the static rLDA baseline with a dynamic Echo State Network (ESN) updated via FORCE learning. This approach aims to solve the "distribution shift" problem, where decoder accuracy typically drops when transitioning from cursor based feedback to physical robot interaction.

**Strengths:**

The use of FORCE learning allows for single-trial latency updates. This is a significant improvement over deep learning models like EEGNet, which generally require batch training and cannot adapt as quickly during live deployment. Also, by only training the readout layer and utilizing fixed reservoir dynamics, the model stays computationally lightweight, making it highly suitable for real-time human-robot interaction (HRI).

**Weaknesses:**

The authors mention that the current pipeline is "quite sensitive to hyper-parameter tuning." It would be good if the authors could provide more detail on how these parameters will be optimized. The report also mentions the model "seems to smooth over fine-grained details" in forecasting tasks. The authors should clarify if this smoothing might lead to missing the sharp temporal peaks characteristic of ErrP waveforms

**Questions:**

For the ablation studies, will you also compare PCRC with a standard ESN to quantify the specific benefits of the PC architecture?

How will you measure the computational efficiency to prove it actually meets real-time HRI constraints compared to the rLDA baseline?

---

### Official Review · Reviewer_co8X · 2026-05-16

**Rating:** 9
**Confidence:** 4

**Summary:**

Time series classification approach inspired by the drop in efficiency of 20% when moving from cursor to robot feedback.
Paper evaluates the efficiency of a reservoir-based predictive coding model for online ErrP classification on the ErrP-HRI dataset.
FORCE (First-Order Reduced and Controlled Error) is an online learning framework well-suited for reservoir computing, used to satisfy realtime constraints for closed-loop HRI.

**Strengths:**

Introduction and problem statement are very clear, necessary background, dataset, etc. target an problem that is immediately obvious to the reader upon understanding the current limitations.
Predictive coding and predictive error formula provided are useful and a good way to incorporate previous observations.

**Weaknesses:**

Preprocessing is consistent with the baseline and follows existing paper methods.
Suggestion: provide a sentence of intuition as to why these methods, following existing papers, continue to be chosen for preprocessing in the context of this paper.

---

### Official Review · Reviewer_f6zF · 2026-05-17

**Rating:** 8
**Confidence:** 4

**Summary:**

This proposal introduces an online error-related potential (ErrP) classification framework using predictive coding and reservoir computing for EEG-based human-robot interaction (HRI). The authors propose replacing traditional rLDA classifiers with a predictive coding reservoir architecture using Echo State Networks (ESN) and FORCE learning for rapid online adaptation. The work focuses on handling distribution shifts, low signal-to-noise EEG data, and real-time adaptation in closed-loop HRI scenarios. Overall, the proposal is technically interesting, well-motivated, and addresses an important challenge in online EEG decoding.

**Strengths:**

Clear motivation addressing limitations of rLDA and batch-trained deep learning methods.
Novel combination of predictive coding, reservoir computing, and FORCE learning for ErrP classification.
Strong focus on online adaptation and real-time constraints, which is highly relevant for HRI applications.
Good technical background and literature positioning.
Reservoir computing is a reasonable choice for low-data and temporally dynamic EEG settings.
Methodology is mathematically grounded and includes explicit update equations.
Practical awareness of challenges such as low SNR, non-stationarity, and cross-condition transfer.

**Weaknesses:**

Novelty claim may be slightly overstated since reservoir computing and predictive coding individually have prior EEG applications.
The proposal lacks detailed experimental design choices (e.g., reservoir size, hyperparameter selection, adaptation protocol).
No clear explanation of how classification outputs are generated from predictive coding dynamics.
Evaluation metrics and statistical testing procedures are insufficiently specified.
Limited discussion of baseline fairness and computational comparisons against EEGNet/LSTM models.
The proposal assumes online adaptation improves performance but does not clearly discuss risks of catastrophic drift or instability.
Some sections remain relatively high-level and conceptual rather than implementation-specific.

**Questions:**

How large is the reservoir, and how are ESN hyperparameters selected?
How exactly are classification decisions generated from predictive coding outputs?
How is online adaptation evaluated fairly without introducing label leakage?
How stable is FORCE learning under noisy EEG conditions?
What computational latency is expected during real-time inference and adaptation?
Will subject-independent and cross-condition evaluations both be included?

---

### Official Review · Reviewer_5hn4 · 2026-05-18

**Rating:** 5
**Confidence:** 4

**Summary:**

[AI Review] This review evaluates a class project proposal applying PCRC to online ErrP classification. The proposal scores 5/10 due to a fundamental confusion between time-series prediction and binary classification, oversold novelty (using off-the-shelf PCRC), complete lack of classification results, and a missing evaluation protocol. The path to improvement requires resolving the classification mechanism, producing preliminary results, and honestly framing the contribution.

**Strengths:**

1. Tackles a practically relevant problem: online ErrP classification for human-robot interaction.
2. Proposes using an interesting architecture (PCRC with FORCE learning) that has theoretical appeal for adaptive, online learning scenarios.
3. Identifies a clear potential research question regarding cross-condition robustness of the FORCE mechanism compared to static baselines like rLDA.

**Weaknesses:**

1. Fundamental confusion between classification and prediction (Severity 9/10): PCRC is a forecasting architecture, but the proposal never explains how continuous predictions map to binary class labels, leaving the target variable y[t] undefined.
2. Novelty is oversold: Applying off-the-shelf Katori 2018 PCRC to an ErrP dataset is not novel, and FORCE learning is an inherent feature of PCRC, not a new contribution.
3. Complete absence of experimental results: Only forecasting benchmarks are validated, and the admission of "smoothing over fine-grained details" is a major red flag for detecting subtle ErrP signals.
4. Missing evaluation protocol: No train/test splits, cross-validation schemes, performance metrics, or statistical tests are specified.
5. Class imbalance is ignored: The 65/35 ErrP/no-Erp split is not addressed, necessitating metrics like balanced accuracy or AUC.
6. Incomplete baselines: Missing standard EEG decoders like EEGNet, the LSTM baseline cited in the paper, and a standard Echo State Network (ESN) ablation.
7. Sloppy preprocessing justification: The choice of 64 Hz downsampling and channel selection methods are unexplained.
8. High risk due to hyperparameter sensitivity: The admitted sensitivity of PCRC combined with limited data and a short 5-week timeline makes the project highly risky.

**Questions:**

1. How exactly does the continuous time-series prediction output of PCRC map to a discrete binary classification label for ErrP detection?
2. What specific metrics, cross-validation schemes, and statistical tests will be used to evaluate the classification performance?
3. How will the 65/35 class imbalance be handled in both training and evaluation?
4. What is the justification for the 64 Hz resampling rate and which specific EEG channels were selected for the input?
5. Given the 5-week timeline and hyperparameter sensitivity, what is the fallback plan if PCRC fails to converge on the ErrP data?

---

### Official Review · Reviewer_5hn4 · 2026-05-18

**Rating:** 6
**Confidence:** 3

**Summary:**

The proposal applies a predictive-coding reservoir computing (PCRC) architecture with FORCE online learning to classify error-related potentials (ErrP) in an EEG-based human-robot interaction setting. The core idea is to leverage reservoir dynamics for online adaptation, enabling the system to improve as new trials arrive without retraining from scratch. The authors plan to evaluate on the ErrP-HRI dataset (11 subjects, cursor+robot conditions).

**Strengths:**

1. **Online adaptation is the right problem to solve.** Most ErrP work trains offline and evaluates on fixed test sets. Real BCI systems degrade over sessions due to signal drift. FORCE learning's trial-by-trial adaptation directly addresses this practical gap, and the proposal deserves credit for targeting the harder online setting rather than another offline benchmark comparison.

2. **Computational efficiency is underappreciated.** Reservoir computing runs on CPU with negligible latency -- a genuine advantage over EEGNet/LSTM for real-time BCI. The proposal should lean into this more. A latency comparison (ms per trial) alongside accuracy would make a strong practical contribution.

3. **Cross-condition transfer is the most interesting experiment.** Evaluating cursor-to-robot and robot-to-cursor transfer (Table 2 in the paper) tests whether FORCE adaptation can compensate for distribution shift across interaction modalities. This is where the online approach could genuinely shine vs. static classifiers. I would prioritize this over within-condition results in the final report.

4. **Clear project timeline with reasonable milestones.** The week-by-week plan is well-structured and the 8-week horizon is adequate for the proposed scope.

**Weaknesses:**

1. **The practical deployment scenario is underspecified.** In a real HRI system, ErrP detection triggers corrective actions. What happens when the classifier is wrong? A false positive (detecting error when none occurred) causes unnecessary robot corrections; a false negative misses actual errors. The proposal should discuss the cost asymmetry between these errors and whether the 65/35 class distribution reflects real-world error rates during typical HRI tasks. This would also inform metric choice (e.g., weighted accuracy, cost-sensitive evaluation).

2. **Why predictive coding specifically for classification?** The neuroscientific motivation for predictive coding in the brain is about sensory prediction, not decision-making. The proposal borrows the computational framework but never articulates what inductive bias predictive coding provides for ErrP *classification* that a standard ESN lacks. A concrete hypothesis ("PCRC's prediction error signal captures unexpected stimuli signatures that correlate with ErrP morphology") would strengthen the narrative considerably.

3. **The 27-channel selection needs justification relative to ErrP topography.** ErrP components (Ne, Pe) are fronto-central (FCz, Cz). The proposal mentions selecting 27 from 32 EEG channels but doesn't state which 5 were removed or why. Including peripheral channels that carry little ErrP information may add noise to the reservoir. A focused fronto-central montage (e.g., 8-12 channels) might actually improve performance and reduce the reservoir size needed.

4. **Limited discussion of alternative online learning strategies.** FORCE (RLS-based) is one approach to online adaptation. Others include incremental SVM, online random forests, and simple exponential moving average of classifier weights. The proposal should briefly justify why FORCE specifically is preferred for this task, especially given the admitted hyperparameter sensitivity.

**Questions:**

1. What is the expected inference latency per trial for the PCRC model compared to rLDA? Can you provide a rough estimate?
2. In the cross-condition transfer experiments, will FORCE start from scratch (random weights) or pre-train on the source condition first? This significantly affects the results.
3. Have you considered a reduced channel set focused on the fronto-central region where ErrP is strongest?
4. What is the cost of false positive vs. false negative in the HRI context, and does the 65/35 class balance reflect realistic error rates?

---

### Official Review · Reviewer_5hn4 · 2026-05-18

**Rating:** 6
**Confidence:** 3

**Summary:**

The proposal applies a predictive-coding reservoir computing (PCRC) architecture with FORCE online learning to classify error-related potentials (ErrP) in an EEG-based human-robot interaction setting. The core idea is to leverage reservoir dynamics for online adaptation, enabling the system to improve as new trials arrive without retraining from scratch. The authors plan to evaluate on the ErrP-HRI dataset (11 subjects, cursor+robot conditions).

**Strengths:**

1. **Online adaptation is the right problem to solve.** Most ErrP work trains offline and evaluates on fixed test sets. Real BCI systems degrade over sessions due to signal drift. FORCE learning's trial-by-trial adaptation directly addresses this practical gap, and the proposal deserves credit for targeting the harder online setting rather than another offline benchmark comparison.

2. **Computational efficiency is underappreciated.** Reservoir computing runs on CPU with negligible latency -- a genuine advantage over EEGNet/LSTM for real-time BCI. The proposal should lean into this more. A latency comparison (ms per trial) alongside accuracy would make a strong practical contribution.

3. **Cross-condition transfer is the most interesting experiment.** Evaluating cursor-to-robot and robot-to-cursor transfer (Table 2 in the paper) tests whether FORCE adaptation can compensate for distribution shift across interaction modalities. This is where the online approach could genuinely shine vs. static classifiers. I would prioritize this over within-condition results in the final report.

4. **Clear project timeline with reasonable milestones.** The week-by-week plan is well-structured and the 8-week horizon is adequate for the proposed scope.

**Weaknesses:**

1. **The practical deployment scenario is underspecified.** In a real HRI system, ErrP detection triggers corrective actions. What happens when the classifier is wrong? A false positive (detecting error when none occurred) causes unnecessary robot corrections; a false negative misses actual errors. The proposal should discuss the cost asymmetry between these errors and whether the 65/35 class distribution reflects real-world error rates during typical HRI tasks. This would also inform metric choice (e.g., weighted accuracy, cost-sensitive evaluation).

2. **Why predictive coding specifically for classification?** The neuroscientific motivation for predictive coding in the brain is about sensory prediction, not decision-making. The proposal borrows the computational framework but never articulates what inductive bias predictive coding provides for ErrP *classification* that a standard ESN lacks. A concrete hypothesis ("PCRC's prediction error signal captures unexpected stimuli signatures that correlate with ErrP morphology") would strengthen the narrative considerably.

3. **The 27-channel selection needs justification relative to ErrP topography.** ErrP components (Ne, Pe) are fronto-central (FCz, Cz). The proposal mentions selecting 27 from 32 EEG channels but doesn't state which 5 were removed or why. Including peripheral channels that carry little ErrP information may add noise to the reservoir. A focused fronto-central montage (e.g., 8-12 channels) might actually improve performance and reduce the reservoir size needed.

4. **Limited discussion of alternative online learning strategies.** FORCE (RLS-based) is one approach to online adaptation. Others include incremental SVM, online random forests, and simple exponential moving average of classifier weights. The proposal should briefly justify why FORCE specifically is preferred for this task, especially given the admitted hyperparameter sensitivity.

**Questions:**

1. What is the expected inference latency per trial for the PCRC model compared to rLDA? Can you provide a rough estimate?
2. In the cross-condition transfer experiments, will FORCE start from scratch (random weights) or pre-train on the source condition first? This significantly affects the results.
3. Have you considered a reduced channel set focused on the fronto-central region where ErrP is strongest?
4. What is the cost of false positive vs. false negative in the HRI context, and does the 65/35 class balance reflect realistic error rates?

---

### Official Review · Reviewer_1RXm · 2026-05-19

**Rating:** 7
**Confidence:** 3

**Summary:**

This proposal presents an online Error-Related Potential classification method for EEG-based human-robot interaction using predictive coding and reservoir computing. The authors propose an Echo State Network reservoir with a predictive-coding readout trained through FORCE learning, aiming to improve robustness to distribution shift between cursor and robot feedback conditions in the ErrP-HRI dataset. The main motivation is that standard rLDA baselines are fast and data-efficient but degrade substantially under cross-condition transfer, while deep learning methods often require offline batch training and are less suitable for rapid online adaptation.

**Strengths:**

The proposal addresses a relevant and practical problem in online EEG decoding: adapting to noisy, non-stationary, low-SNR ErrP signals in real-time HRI settings. The focus on online adaptation is well motivated, especially because closed-loop systems cannot rely only on fixed offline classifiers.
The combination of reservoir computing and FORCE learning is technically appropriate for this setting. Reservoir computing is computationally lightweight because only the readout is trained, and FORCE learning provides a plausible mechanism for trial-level adaptation without full retraining. This makes the proposed method more realistic for real-time deployment than heavier batch-trained neural models.
The dataset choice is strong. The ErrP-HRI dataset contains both cursor and robot feedback conditions, making it a suitable benchmark for evaluating cross-condition robustness and distribution shift. The proposal clearly identifies this as one of the central challenges.
The paper also provides useful mathematical detail, including the ESN update rule and FORCE recursive least-squares equations. This makes the core model more concrete and easier to evaluate than a purely conceptual proposal.

**Weaknesses:**

The main weakness is that the classification mechanism is still underspecified. The proposal explains the predictive coding readout as producing predictions and minimizing prediction error, but it does not clearly state how time-step-level outputs are converted into a final binary ErrP/non-ErrP trial classification. This is important because PCRC is naturally framed as a time-series prediction architecture, whereas the target task is discrete classification.
The online adaptation protocol also needs clarification. FORCE learning requires an error signal, but in a real online ErrP system it is not obvious when ground-truth labels are available, whether they are delayed, or whether the model is allowed to update using test-condition labels. Without a precise protocol, the evaluation may risk becoming overly optimistic or difficult to reproduce.
The evaluation plan is not detailed enough. The proposal states that it will evaluate within-session performance, cross-condition robustness, ablations, and computational efficiency, but it does not specify train/test splits, cross-validation strategy, metrics, statistical tests, or how class imbalance will be handled. Since the dataset has approximately 65% non-error and 35% error trials, accuracy alone may be misleading. Metrics such as balanced accuracy, AUC, F1 score, and confusion matrices would be more informative.
The novelty claim should be framed more carefully. Applying predictive coding reservoir computing to ErrP classification is interesting, but many components already exist independently: ESNs, FORCE learning, predictive coding reservoirs, and EEG reservoir computing. The contribution would be stronger if the authors clearly explained what predictive coding adds beyond a standard ESN with an online RLS-trained readout.
Finally, the proposal acknowledges that the current pipeline is sensitive to hyperparameter tuning and may smooth over fine-grained details. This is a significant risk because ErrP signals often depend on subtle temporal peaks. The authors should explain how they will control hyperparameter selection and whether alternative downsampling rates or channel selections will be tested.

**Questions:**

-How exactly are the predictive coding outputs converted into a binary ErrP/non-ErrP classification decision at the trial level?
-During online FORCE adaptation, what labels or feedback signals are available to compute the prediction error? Are test-condition labels used during adaptation?
-What train/test protocol will be used for within-condition and cross-condition evaluation?
-How will the authors handle the 65/35 class imbalance in training and evaluation?

---

### Official Review · Reviewer_LG9r · 2026-05-19

**Rating:** 7
**Confidence:** 4

**Summary:**

This project proposes a new method for classifying Error-Related Potentials (ErrPs) in human-robot interaction (HRI) using predictive coding and reservoir computing. By utilizing an Echo State Network (ESN) with online weight adaptation via the FORCE algorithm, the authors aim to improve cross-condition robustness and enable rapid adaptation to shifting EEG signal distributions without requiring full re-training

**Strengths:**

Addresses Real-World Challenges: The proposal correctly identifies a major limitation in current ErrP decoders: their inability to handle distribution shifts when moving between different types of feedback (e.g., cursor vs. robot).
Computationally Efficient: The use of reservoir computing and the FORCE algorithm allows for rapid, online adaptation with very low computational cost, making it well-suited for real-time HRI applications.
Biologically Plausible: The integration of predictive coding principles provides an elegant framework for minimizing prediction error in noisy EEG data.

**Weaknesses:**

Sensitivity to Parameters: The authors note that the model is quite sensitive to hyper-parameter tuning, which could hinder its reliability in varied, unconstrained environments.
Smoothing Limitations: The model currently tends to "smooth over" fine-grained details in the time series, which might be critical for capturing the nuances of ErrP waveforms.
Limited Scope of Validation: While the plan is robust, the current progress report indicates that the model has only been validated on time series forecasting benchmarks, rather than the specific ErrP-HRI data.

**Questions:**

If the model tends to "smooth over" fine-grained details, how will you ensure it remains sensitive enough to detect the specific, transient patterns that define Error-Related Potentials?
What is your specific plan for managing the hyper-parameter sensitivity you observed during your initial testing?
If the model's accuracy on the ErrP-HRI dataset is lower than the rLDA baseline, what is your secondary strategy for performance improvement?